# Mathematical Modelling of the Impact of Non-Pharmacological Strategies to Control the COVID-19 Epidemic in Portugal

**Constantino Caetano** [1,*], **Maria Luísa Morgado** [2,3], **Paula Patrício** [4], **João F. Pereira** [3] and **Baltazar Nunes** [1,5]

1   Instituto Nacional de Saúde Doutor Ricardo Jorge, 1649-016 Lisbon, Portugal; Baltazar.Nunes@insa.min-saude.pt
2   Center for Computational and Stochastic Mathematics, Instituto Superior Técnico, University of Lisbon, 1049-001 Lisbon, Portugal; luisam@utad.pt
3   Department of Mathematics, University of Trás-os-Montes e Alto Douro, UTAD, 5001-801 Vila Real, Portugal; pereira.jpf96@gmail.com
4   Center for Mathematics and Applications (CMA), FCT NOVA and Department of Mathematics, FCT NOVA, Quinta da Torre, 2829-516 Caparica, Portugal; pcpr@fct.unl.pt
5   Centro de Investigação em Saúde Pública, Escola Nacional de Saúde Pública, Universidade NOVA de Lisboa, 1600-560 Lisbon, Portugal
*   Correspondence: constantino.caetano@insa.min-saude.pt; Tel.: +351-21-751-9200

**Abstract:** In this paper, we present an age-structured SEIR model that uses contact patterns to reflect the physical distance measures implemented in Portugal to control the COVID-19 pandemic. By using these matrices and proper estimates for the parameters in the model, we were able to ascertain the impact of mitigation strategies employed in the past. Results show that the March 2020 lockdown had an impact on disease transmission, bringing the effective reproduction number ($\mathcal{R}(t)$) below 1. We estimate that there was an increase in the transmission after the initial lift of the measures on 6 May 2020 that resulted in a second wave that was curbed by the October and November measures. December 2020 saw an increase in the transmission reaching an $\mathcal{R}(t) = 1.45$ in early January 2021. Simulations indicate that the lockdown imposed on the 15 January 2021 might reduce the intensive care unit (ICU) demand to below 200 cases in early April if it lasts at least 2 months. As it stands, the model was capable of projecting the number of individuals in each infection phase for each age group and moment in time.

**Keywords:** epidemiological models; SEIR type compartmental model; COVID-19; mathematical modelling; contact matrices

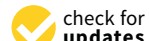

## 1. Introduction

On the 11th of March 2020, the World Health Organisation declared the global public health emergency epidemic of COVID-19 a pandemic. The first confirmed case in Portugal occurred on 2nd of March 2020. During the following days, the number of infections increased significantly, a fact that led the policy makers to act. Schools closed on the 16th March, followed by the implementation of a state of emergency on the 22nd. This lockdown severely restricted movement of individuals within the country as well as instating a mandatory stay-at-home order for the population. Some exceptions were made for individuals working in industries that were essential. Borders had restrictions but were not closed.

This lockdown led to a decline in the incidence of SARS-CoV-2 infection cases from early April until late May, when some of the mitigation measures were lifted, and the number of infections stabilised. The end of summer led to a resurgence in cases, which happened throughout Europe, and the number of new cases has risen ever since, having only stabilised in the last few days of November. Portugal entered into a state of calamity on the 15th of October, followed by the mandatory use of face masks outdoors on the 4th

of November and county-specific measures, depending of the 14-day cumulative incidence in each county.

Since the appearance of the first cases in Wuhan, China, in late December of 2019, several research teams have employed the use of mathematical and statistical techniques to ascertain the course of the disease spread. Common mathematical tools to describe such phenomena are systems of ordinary differential equations. The most notable are the SIR and SEIR models, first developed by Kermack and McKendrick (1927). Throughout the course of the COVID-19 pandemic, these models have been adapted from the SEIR model to study an array of different epidemic questions. During the early months, these models were employed to nowcast and forecast the national spread of SARS-CoV-2 [1]. Similar models were also used to estimate the case ascertainment rate [2]. This is of note regarding the transmission dynamic since it has been shown that a percentage of infected individuals do not develop symptoms [3], but are still capable of infecting others [4]. One of the main purposes of these modelling techniques is to evaluate the impact of contagion mitigation measures, such as the closure of schools and lockdowns [5].

In this work, we use a SEIR-type model that takes into account the contact mixing between different age groups depending on contact patterns at work, home, school and other locations. Furthermore, it includes asymptomatic transmission, hospitalisation, recovery and death. Contact tracing with isolation and isolation compliance and detection were also implemented in the model framework. Upon proper parameter estimation, it is possible to simulate the effect of non-pharmaceutical-interventions (NPI), such as the closure of schools, mask usage, shielding of elderly individuals, reducing effective contacts in workplaces, on the number of hospitalised individuals, or individuals who require intensive care. Additionally, we were also able to estimate the effect that the NPIs implemented up to 15th of January 2021 had on the disease transmission.

The paper is organised as follows: Section 2 pertains to the model description and model analysis. Here we start by describing the model's architecture. We explain each compartment and respective flows. We also present the system of ordinary differential equations. In this section, we consider two types of models: the homogeneous model and the heterogeneous model. The former represents the model with random mixing, the latter pertains to the model with heterogeneous mixing among age-groups. Reproduction numbers are derived in this chapter for each model. In Section 3, we detail the data used to fit the model. We present estimates for each fixed parameter and their sources. Results are presented in Section 4, where we detail the fitting of the model and estimate the effect that the NPIs implemented so far had on the disease transmission and create scenario simulations to forecast the effect of the lockdown measures instated on the 15th of January 2021. In Section 5 we discuss the model results, its strengths and limitations.

## 2. Materials and Methods

### 2.1. Model Description

In our model formulation we assumed Portugal to be a closed system with constant population during the epidemic. An individual starts off as susceptible ($S$). After becoming infected, but not yet infectious, it can either move into the exposed traced ($E_t$), or exposed non traced compartment ($E_{nt}$), that is, individuals that have been traced and individuals not caught in contact tracing, respectively. Considering the homogeneous model, that is, the case where no age stratification is considered, individuals become $E_t$ at a rate $q\lambda$, where $\lambda$ is the force of infection and $q$ represents the proportion of individuals that are identified in the contact tracing process and are isolated.

An infected individual becomes infectious at $\varepsilon$ rate. These individuals become either asymptomatic ($I_A$ and $I_{A_q}$) with probability $p$ or symptomatic ($I_S$ and $I_{S_q}$).

Isolated individuals are assumed to have reduced contacts during their infectious period. Some of the individuals in the latter stage might refuse to comply with isolation demands, hence a fraction $(1 - c)$ of the traced individuals progress to non isolation infectious compartments $I_S$ or $I_A$. A proportion ($d$) of infected individuals whom are not

isolated, might decide to self-isolate before they become infectious. This behavior might stem from an individual's risk perception.

Asymptomatic individuals infect others at a reduced rate $\alpha_A$, although not negligible, since it has been shown that asymptomatic transmission makes up a large portion of infections [4]. Asymptomatic and symptomatic individuals are removed at $r_a$ and $r_s$ rates, respectively. A proportion of symptomatic individuals, $\theta$, will develop a more serious form of the disease which will require hospitalisation ($H$). Asymptomatic individuals are assumed to always recover from the disease. Hospitalised individuals are removed at a $\rho$ rate. These individuals can either require critical care ($H_{ICU}$) with probability $\pi$, die ($M$) with probability $\tau$ or recover ($R$ and $R_{obs}$). Immunity to reinfection is assumed for the time window studied here (beginning February 2020 until mid January 2021).

$$S' = -\lambda S, \tag{1}$$
$$E'_t = q\lambda S - \varepsilon E_t, \tag{2}$$
$$I'_A = p\varepsilon E_{nt} + (1-c)p\varepsilon E_t - r_a I_A, \tag{3}$$
$$I'_S = (1-d)(1-p)\varepsilon E_{nt} + (1-c)(1-p)\varepsilon E_t - r_s I_S, \tag{4}$$
$$E'_{nt} = (1-q)\lambda S - \varepsilon E_{nt}, \tag{5}$$
$$I'_{Aq} = cp\varepsilon E_t - r_a I_{Aq}, \tag{6}$$
$$I'_{Sq} = c(1-p)\varepsilon E_t + d(1-p)\varepsilon E_{nt} - r_s I_{Sq}, \tag{7}$$
$$H' = \theta r_s(I_S + I_{Sq}) - \rho H, \tag{8}$$
$$H'_{ICU} = \pi\rho H - \omega H_{ICU}, \tag{9}$$
$$M' = \mu\omega H_{ICU} + \tau\rho H, \tag{10}$$
$$R'_{obs} = (1-\theta)r_s I_{Sq} + r_a I_{Aq} + (1-\pi-\tau)\rho H + (1-\mu)\omega H_{ICU}, \tag{11}$$
$$R' = (1-\theta)r_s I_S + r_a I_A. \tag{12}$$

In the homogeneous model case, that is, the case where the population is not divided into age groups and random mixing is assumed, the force of infection is given as $\lambda S = \beta(\alpha_S C_S I_S + \alpha_A C_A I_A + \alpha_{Sq} C_{Sq} I_{Sq} + \alpha_{Aq} C_{Aq} I_{Aq})\frac{S}{N}$, where $\alpha_i$ and $C_i$, for $i = S, A, Sq, Aq$, are the relative transmission and contacts of infectious classes $I_A$, $I_{A_q}$, $I_S$ and $I_{S_q}$.

A schematic diagram of the transmission model is represented in Figure 1.

For the heterogeneous version of the model we divide each compartment in $n$ age classes. Later on we discuss the possibility of parameters to be considered age dependent. The force of infection of age-group $j$ onto age-group $i$ is:

$$S_i\lambda_{ij} = \beta S_i\left[\alpha_S \mathfrak{C}^S_{ij} I_{Sj}/N_j + \alpha_A \mathfrak{C}^A_{ij} I_{Aj}/N_j + \alpha_{Sq} \mathfrak{C}^{Sq}_{ij} I_{Sqj}/N_j + \alpha_{Aq} \mathfrak{C}^{Aq}_{ij} I_{Aqj}/N_j\right],$$

where $\alpha_k$ and $\mathfrak{C}^k_{ij}$, for $k = S, A, Sq, Aq$, $i,j = 1,...,n$, are the relative transmission and contact matrices of infectious classes. Each contact matrix is of the form:

$$\mathfrak{C}^k = U^k C^w + W^k C^{sch} + Y^k C^h + Z^k C^o, \tag{13}$$

with $U^k$, $W^k$, $Y^k$ and $Z^k$, for $k = S, A, Sq, Aq$, are $n \times n$ diagonal matrices with positive entries which are used to change the number of contacts in each location and between age groups, and $C^l$, for $l = w, sch, h, o$, correspond to contact matrices at work (w), school (sch), home (h) and other social contacts (o). Hence if we consider $n$ age-groups the force of infection exerted onto age-group $i$ is described as:

$$\lambda_{i\cdot} = \beta \sum_{j=1}^{n} \alpha_S \mathfrak{C}^S_{ij} I_{Sj}/N_j + \alpha_A \mathfrak{C}^A_{ij} I_{Aj}/N_j + \alpha_{Sq} \mathfrak{C}^{Sq}_{ij} I_{Sqj}/N_j + \alpha_{Aq} \mathfrak{C}^{Aq}_{ij} I_{Aqj}/N_j,$$

meaning that individuals in age-group $i$ become infected depending on their contacts with all the age groups [6].

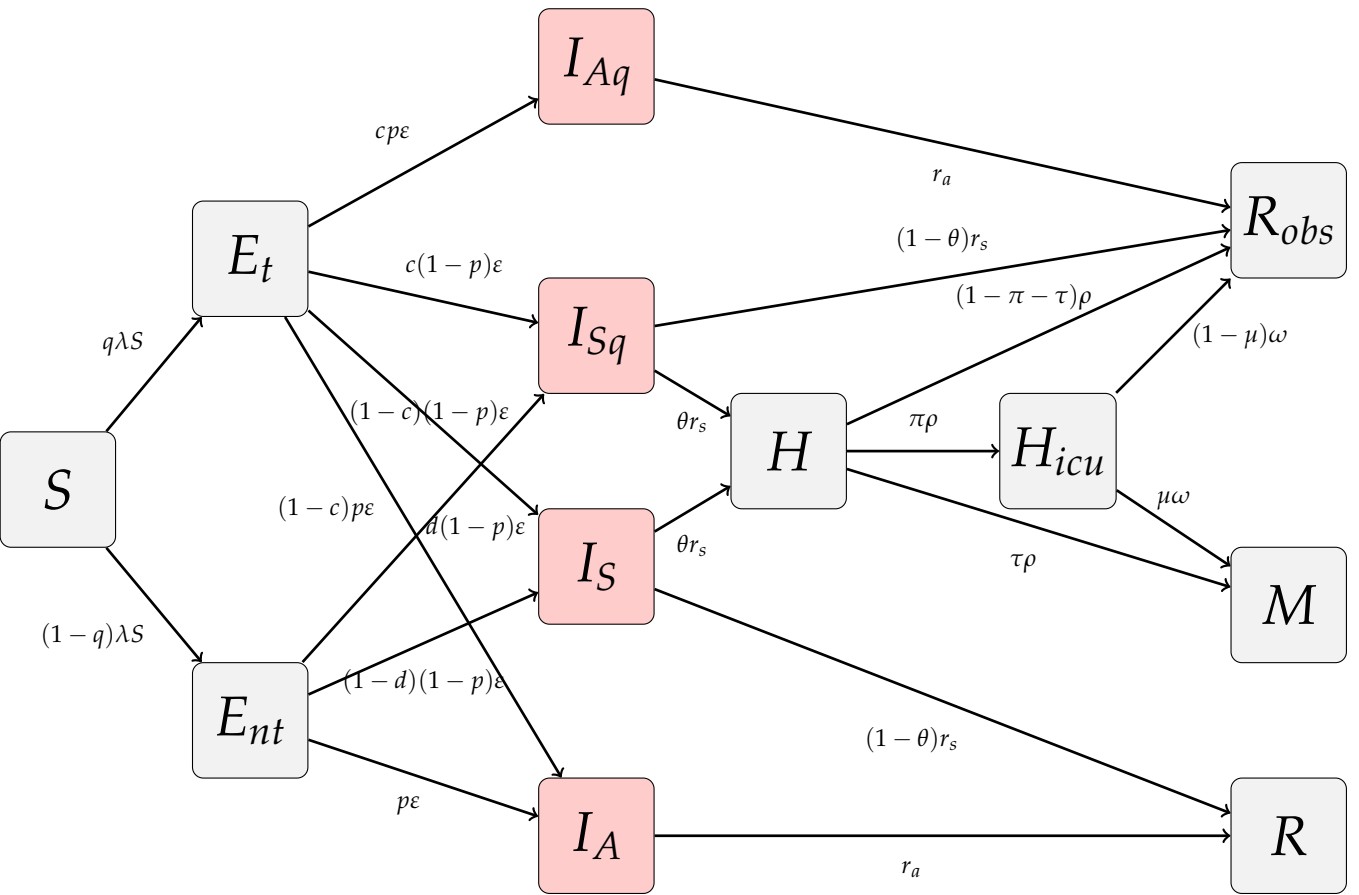

**Figure 1.** Schematic diagram of the homogeneous COVID-19 transmission model. Disease transmission is $\lambda S = \beta(\alpha_S C_S I_S + \alpha_A C_A I_A + \alpha_{Sq} C_{Sq} I_{Sq} + \alpha_{Aq} C_{Aq} I_{Aq}) \frac{S}{N}$, where $\alpha_i$ and $C_i$, for $i = S, A, Sq, Aq$, are the relative transmissibility and contacts of infectious classes.

*2.2. Model Analysis*

2.2.1. Homogeneous Model

Our compartmental model for COVID-19 infection transmission is a 12 equation system (1)–(12). It can be seen as three variables system assuming $s(t) = S$ a source variable, $i(t) = [E_{nt}\ I_A\ I_S\ E_t\ I_{Aq}\ I_{Aq}\ H\ H_{ICU}]^T$ a transient variable and $r(t) = [M\ R\ R_{obs}]^T$ a cumulative variable, as in a typical SIR epidemic model.

The population is kept constant over the time, since it verifies $S'(t) + E'_{nt}(t) + I'_A(t) + I'_S(t) + E'_t(t) + I'_{Aq}(t) + I'_{Sq}(t) + H'(t) + H'_{ICU}(t) + M'(t) + R'_{0bs}(t) + R'(t) = N$, where $N$ is the total population.

Given an initial condition $S(0), E_{nt}(0), I_A(0), I_S(0), E_t(0), I_{Aq}(0), I_{Sq}(0), H(0), H_{ICU}(0),$ $M(0), R_{obs}(0), R(0) \geq 0$, summing to $N$, all possible equilibria are disease free of the form $(\overline{S}, 0, 0, 0, 0, 0, 0, 0, 0, \overline{M}, \overline{R}, \overline{R}_{obs})$, meaning that $\overline{S} \geq 0, \overline{E}_{nt} = 0, \overline{I}_A = 0, \overline{I}_S = 0, \overline{E}_t = 0,$ $\overline{I}_{Aq} = 0, \overline{I}_{Sq} = 0, \overline{H} = 0, \overline{H}_{ICU} = 0, \overline{M} \geq 0, \overline{R}_{obs} \geq 0$ and $\overline{R} \geq 0$, with $\overline{S} + \overline{M} + \overline{R} + \overline{R}_{obs} = N$.

We first study the asymptotic behavior of the model and linearise system (1)–(12) around the disease-free steady state. Following the *next generation approach* in [7], the resulting linearised sub-system for the 6 infected classes is of the form

$$X' = (F - V)X, \text{ where } X = [E_{nt}\ I_A\ I_S\ E_t\ I_{Aq}\ I_{Sq}]^T,$$

where the $F$ matrix corresponds to new infections:

$$F = \begin{bmatrix} 0 & q\beta\alpha_{Aq}C_{Aq}\frac{\bar{S}}{N} & q\beta\alpha_{Sq}C_{Sq}\frac{\bar{S}}{N} & 0 & q\beta\alpha_A C_A\frac{\bar{S}}{N} & q\beta\alpha_S C_S\frac{\bar{S}}{N} \\ 0 & 0 & 0 & 0 & 0 & 0 \\ 0 & 0 & 0 & 0 & 0 & 0 \\ 0 & (1-q)\beta\alpha_{Aq}C_{Aq}\frac{\bar{S}}{N} & (1-q)\beta\alpha_{Sq}C_{Sq}\frac{\bar{S}}{N} & 0 & (1-q)\beta\alpha_A C_A\frac{\bar{S}}{N} & (1-q)\beta\alpha_S C_S\frac{\bar{S}}{N} \\ 0 & 0 & 0 & 0 & 0 & 0 \\ 0 & 0 & 0 & 0 & 0 & 0 \end{bmatrix}, \tag{14}$$

and $V$ describes the other transitions between compartments

$$V = \begin{bmatrix} \varepsilon & 0 & 0 & 0 & 0 & 0 \\ -cp\varepsilon & r_a & 0 & 0 & 0 & 0 \\ -c(1-p)\varepsilon & 0 & rs & -d(1-p)\varepsilon & 0 & 0 \\ 0 & 0 & 0 & \varepsilon & 0 & 0 \\ -(1-c)p\varepsilon & 0 & 0 & -p\varepsilon & r_a & 0 \\ -(1-c)(1-p)\varepsilon & 0 & 0 & -(1-d)(1-p)\varepsilon & 0 & r_s \end{bmatrix}. \tag{15}$$

The next generation matrix, $FV^{-1}$, is

$$\beta\frac{\bar{S}}{N}\begin{bmatrix} q\mathcal{A} & q\frac{\alpha_{Aq}}{r_a}C_{Aq} & q\frac{\alpha_{Sq}}{r_s}C_{Sq} & q\mathcal{B} & q\frac{\alpha_A}{r_a}C_A & q\frac{\alpha_S}{r_s}C_S \\ 0 & 0 & 0 & 0 & 0 & 0 \\ 0 & 0 & 0 & 0 & 0 & 0 \\ (1-q)\mathcal{A} & (1-q)\frac{\alpha_{Aq}}{r_a}C_{Aq} & (1-q)\frac{\alpha_{Sq}}{r_s}C_{Sq} & (1-q)\mathcal{B} & (1-q)\frac{\alpha_A}{r_a}C_A & (1-q)\frac{\alpha_S}{r_s}C_S \\ 0 & 0 & 0 & 0 & 0 & 0 \\ 0 & 0 & 0 & 0 & 0 & 0 \end{bmatrix}, \tag{16}$$

where

$$\mathcal{A} = \frac{(1-c)p\alpha_A}{r_a}C_A + \frac{cp\alpha_{Aq}}{r_a}C_{Aq} + \frac{(1-c)(1-p)\alpha_S}{r_s}C_S + \frac{c(1-p)\alpha_{Sq}}{r_s})C_{Sq}$$

and

$$\mathcal{B} = \frac{p\alpha_A}{r_a}C_A + \frac{(1-d)(1-p)\alpha_S}{r_s}C_S + \frac{d(1-p)\alpha_{Sq}}{r_s}C_{Sq}.$$

The reproduction number is then the spectral radius of the next generation matrix $FV^{-1}$

$$\begin{aligned} \mathcal{R} = \rho(FV^{-1}) &= \beta\left[\frac{p\alpha_A C_A}{r_a}((1-q)+q(1-c)) + \frac{p\alpha_{Aq}C_{Aq}}{r_a}qc+\right. \\ &\left.+\frac{(1-p)\alpha_S C_S}{r_s}((1-c)q+(1-q)(1-d)) + \frac{(1-p)\alpha_{Sq}C_{Sq}}{r_s}(qc+d(1-q))\right]\frac{\bar{S}}{N} \\ &= \beta\left[\frac{p}{r_a}[((1-q)+q(1-c))\alpha_A C_A + qc\alpha_{Aq}C_{Aq}]+\right. \\ &\left.+\frac{1-p}{r_s}[(q(1-c)+(1-q)(1-d))\alpha_S C_S + (qc+d(1-q))\alpha_{Sq}C_{Sq}]\right]\frac{\bar{S}}{N} \\ &= R_0\frac{\bar{S}}{N}. \end{aligned} \tag{17}$$

Hence, for

$$R_0 \frac{\overline{S}}{N} < 1, \tag{18}$$

the disease free equilibrium is locally asymptotically stable and it is unstable otherwise. Based on the linearized system we also define a time dependent effective reproduction number as:

$$\mathcal{R}(t) = R_0 \frac{S(t)}{N}, \tag{19}$$

where $R_0$ can be obtained from (17).

Finally, the basic reproduction number of this system, assumed as the average number of infected cases produced by one infected individual during the infectious period assuming a fully susceptible population ($\overline{S} = N$) and $q = 0$ is then

$$
\begin{aligned}
\mathcal{R}_0 &= \beta \left[ \frac{p\alpha_A C_A}{r_a} + \frac{(1-d)(1-p)\alpha_S C_S}{r_s} + \frac{d(1-p)\alpha_{Sq} C_{Sq}}{r_s} \right] \\
&= \beta \left[ \frac{p\alpha_A C_A}{r_a} + \frac{(1-p)}{r_s}\left((1-d)\alpha_S C_S + d\alpha_{Sq} C_{Sq}\right) \right].
\end{aligned}
\tag{20}
$$

### 2.2.2. Heterogeneous Model

In the heterogeneous version of our COVID-19 model, we consider that each compartment is divided into $n$ age classes to accommodate differences in contacts. We can also assume other disease related parameters depending on age, further details will be discussed in Section 3.

Once more, all possible equilibria are disease free of the form $(\overline{S}_i, 0, 0, 0, 0, 0, 0, 0, 0, \overline{M}_i, \overline{R}_i, \overline{R}_{obs_i})$, with $\overline{S}_i + \overline{M}_i + \overline{R}_i + \overline{R}_{obs_i} = N_i$ and

$$\sum_i^n (\overline{S} + \overline{M}_i + \overline{R}_i + \overline{R}_{obs})_i = N,$$

where $N_i$ is the population size of age-group $i$.

We first study the asymptotic behavior of the model and linearize system (1–12) around the disease-free steady state. Following the next generation approach as previously we construct matrices $F$ and $V$. $F$ is a $n \times n$ block matrix where each block is of the form:

$$
F_{ij} = \begin{bmatrix}
0 & q\beta\alpha_{Aq}\mathfrak{C}_{ij}^{Aq}\frac{\overline{S}_i}{N_j} & q\beta\alpha_{Sq}\mathfrak{C}_{ij}^{Sq}\frac{\overline{S}_i}{N_j} & 0 & q\beta\alpha_A\mathfrak{C}_{ij}^{A}\frac{\overline{S}_i}{N_j} & q\beta\alpha_S\mathfrak{C}_{ij}^{S}\frac{\overline{S}_i}{N_j} \\
0 & 0 & 0 & 0 & 0 & 0 \\
0 & 0 & 0 & 0 & 0 & 0 \\
0 & (1-q)\beta\alpha_{Aq}\mathfrak{C}_{ij}^{Aq}\frac{\overline{S}_i}{N_j} & (1-q)\beta\alpha_{Sq}\mathfrak{C}_{ij}^{Sq}\frac{\overline{S}_i}{N_j} & 0 & (1-q)\beta\alpha_A\mathfrak{C}_{ij}^{A}\frac{\overline{S}_i}{N_j} & (1-q)\beta\alpha_S\mathfrak{C}_{ij}^{S}\frac{\overline{S}_i}{N_j} \\
0 & 0 & 0 & 0 & 0 & 0 \\
0 & 0 & 0 & 0 & 0 & 0
\end{bmatrix}, \tag{21}
$$

where $\overline{S}_i$ is the susceptible at the disease free equilibrium considered of age class $i$, $N_j$ is the total population of age class $j$, with $i, j = 1, \ldots, n$ and $n$ is the number of age classes considered. $V$ is a diagonal $n$ block matrix where each block if of the form:

$$
V = \begin{bmatrix}
\varepsilon & 0 & 0 & 0 & 0 & 0 \\
-cp\varepsilon & r_a & 0 & 0 & 0 & 0 \\
-c(1-p)\varepsilon & 0 & rs & -d(1-p)\varepsilon & 0 & 0 \\
0 & 0 & 0 & \varepsilon & 0 & 0 \\
-(1-c)p\varepsilon & 0 & 0 & -p\varepsilon & r_a & 0 \\
-(1-c)(1-p)\varepsilon & 0 & 0 & -(1-d)(1-p)\varepsilon & 0 & r_s
\end{bmatrix}. \tag{22}
$$

Hence, each $6 \times 6$ block of the next generation matrix $FV^{-1}$, $(FV^{-1})_{ij}$ has the form:

$$
\beta \begin{bmatrix}
q\mathcal{A} & q\frac{\alpha_{Aq}}{r_a}\tilde{\mathfrak{C}}^{Aq}_{ij} & q\frac{\alpha_{Sq}}{r_s}\tilde{\mathfrak{C}}^{Sq}_{ij} & q\mathcal{B} & q\frac{\alpha_A}{r_a}\tilde{\mathfrak{C}}^{A}_{ij} & q\frac{\alpha_S}{r_s}\tilde{\mathfrak{C}}^{S}_{ij} \\
0 & 0 & 0 & 0 & 0 & 0 \\
0 & 0 & 0 & 0 & 0 & 0 \\
(1-q)\mathcal{A} & (1-q)\frac{\alpha_{Aq}}{r_a}\tilde{\mathfrak{C}}^{Aq}_{ij} & (1-q)\frac{\alpha_{Sq}}{r_s}\tilde{\mathfrak{C}}^{Sq}_{ij} & (1-q)\mathcal{B} & (1-q)\frac{\alpha_A}{r_a}\tilde{\mathfrak{C}}^{A}_{ij} & (1-q)\frac{\alpha_S}{r_s}\tilde{\mathfrak{C}}^{S}_{ij} \\
0 & 0 & 0 & 0 & 0 & 0 \\
0 & 0 & 0 & 0 & 0 & 0
\end{bmatrix},
\tag{23}
$$

where

$$
\mathcal{A} = \frac{(1-c)p\alpha_A}{r_a}\tilde{\mathfrak{C}}^{A}_{ij} + \frac{cp\alpha_{Aq}}{r_a}\tilde{\mathfrak{C}}^{Aq}_{ij} + \frac{(1-c)(1-p)\alpha_S}{r_s}\tilde{\mathfrak{C}}^{S}_{ij} + \frac{c(1-p)\alpha_{Sq}}{r_s})\tilde{\mathfrak{C}}^{Sq}_{ij}
$$

and

$$
\mathcal{B} = \frac{p\alpha_A}{r_a}\tilde{\mathfrak{C}}^{A}_{ij} + \frac{(1-d)(1-p)\alpha_S}{r_s}\tilde{\mathfrak{C}}^{S}_{ij} + \frac{d(1-p)\alpha_{Sq}}{r_s}\tilde{\mathfrak{C}}^{Sq}_{ij},
$$

for $\tilde{\mathfrak{C}}^{k}_{ij} = \mathfrak{C}^{k}_{ij}\frac{\overline{S}_i}{N_j}$, $k = A, Aq, S, Sq$. The basic reproduction number ($q = 0, c = 1$) and the reproduction number for the complete heterogeneous model are then the spectral radius of the corresponding next generation matrices $FV^{-1}$.

If we assume that matrix $V$ does not depend on the age classes we can get more detailed expressions due to the block structure of the matrices

$$
\mathcal{R}^{het}_0 = \rho(FV^{-1}) = \beta\rho\left(\left[\frac{p\alpha_A}{r_a}\tilde{\mathfrak{C}}^{A}_{ij} + \frac{(1-d)(1-p)\alpha_S}{r_s}\tilde{\mathfrak{C}}^{S}_{ij} + \frac{d(1-p)\alpha_{Sq}}{r_s}\tilde{\mathfrak{C}}^{Sq}_{ij}\right]\right)
\tag{24}
$$

and in order to compute the effective reproduction number, whenever $q \neq 0$, we obtain:

$$
R^{het}_0 = \rho(FV^{-1}) = \beta\,\rho\left(\left[\frac{p\alpha_A}{r_a}((1-q) + q(1-c))\,\tilde{\mathfrak{C}}^{A}_{ij} + \frac{p\alpha_{Aq}}{r_a}qc\,\tilde{\mathfrak{C}}^{Aq}_{ij} + \right.\right.
$$
$$
\left.\left. + \frac{(1-p)\alpha_S}{r_s}((1-c)q + (1-d)(1-q))\,\tilde{\mathfrak{C}}^{S}_{ij} + \frac{(1-p)\alpha_{Sq}}{r_s}(qc + d(1-q))\,\tilde{\mathfrak{C}}^{Sq}_{ij}\right]\right).
\tag{25}
$$

## 3. Data

We fixed some disease related parameters according to available data. We used Portuguese data whenever possible. The proportion of asymptomatic individuals was obtained from the first wave of the Portuguese COVID-19 serological survey [3]. The reduced rate at which asymptomatic individuals infected others was obtained in [8]. Data on the duration of the latent and infectious period were assumed to be the same as in [9].

Contact data was obtained from [10], where the authors estimated the Portuguese contact patterns based on the POLYMOD study [11] using a Bayesian hierarchical model and data from health surveys. Figure 2 presents the contact matrices for each setting, which correspond to the matrices $C^h$, $C^{sch}$, $C^w$ and $C^o$ described in (13). Values for the Portuguese population for each age-group was obtained in Statistics Portugal (INE), using projected population for the year of 2019. Data on the mean duration of hospitalisations ($\rho$) and ICU admissions ($\omega$), the proportion of individuals who need intensive care ($\pi$) and the proportion of individuals who die in each hospitalisation category due to COVID-19 ($\tau,\mu$) was obtained in BIMH SPMS/ACSS. The proportion of hospitalised individuals in each age-group was obtained in [12]. The parameter $\beta$ was obtained from (25) by fixing all other parameters and assuming $R_0 = 2.5$ [13]. Parameter values can be found on Appendix A.

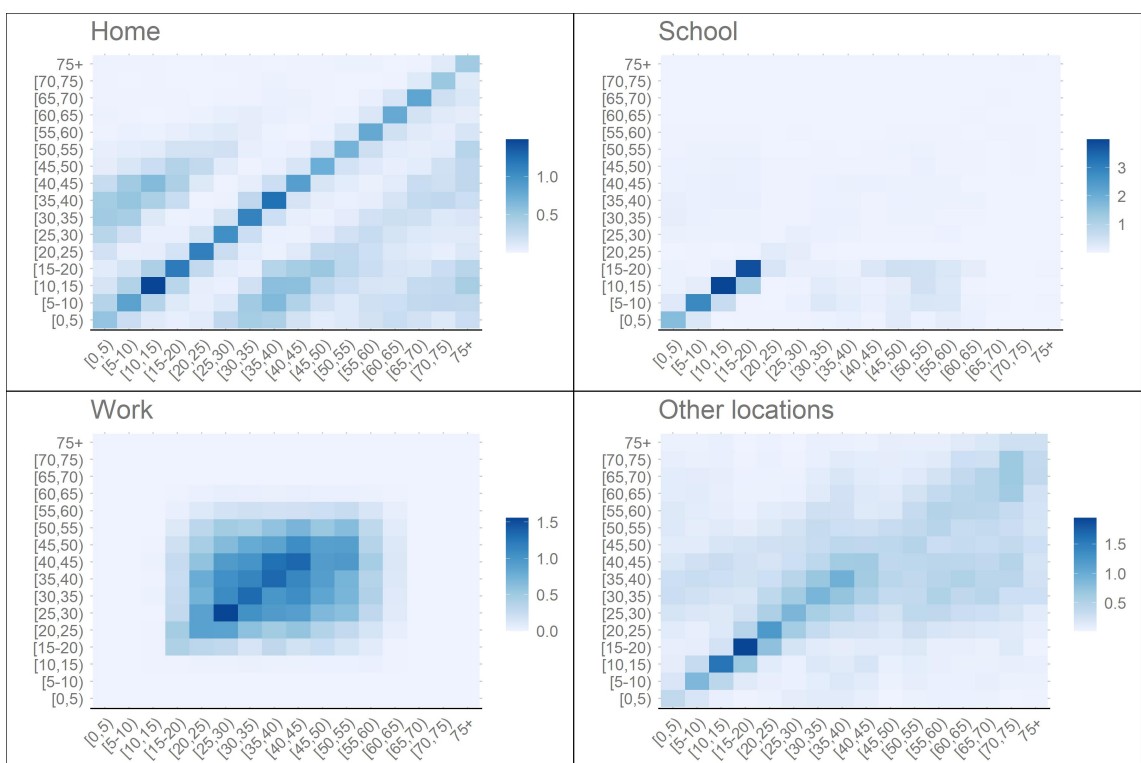

**Figure 2.** Portuguese synthetic contact patterns for school, home, work and other locations prior to the pandemic ([10,11]).

Time series data for hospitalisation, ICU and deaths was extracted each day from the Portuguese General-Directorate of Health COVID-19 online situation report [14].

## 4. Model Fit Strategy, Results and Simulations

The model has been fitted to the time series of the total number of individuals in ICU for each day in Portugal from 10th of February 2020 to the 15th of January 2021. This data was extracted from the Portuguese General-Directorate of Health COVID-19 online situation report [14]. ICU can be considered a more robust indicator than incidence data, since it is not tied to testing strategy and capacity. To our knowledge, in Portugal, the criteria to admit a COVID-19 patient to ICU has not changed. Due to the lack of reliable data, for the present work, we considered a simpler form of the model, by assuming that $q = 0$ and $d = 0$. Future work will address contact trace, compliance and case detection when more detailed data is available.

We assumed that the infectious period, in symptomatic and non-symptomatic individuals, was the same. Day 0 was assumed to be the 10th of February 2020, since it was an incubation period (approximately 5 days), [15], prior to the first disease onset in Portugal [16]. Initial conditions were estimated for the first influx of infected but not yet infectious individuals. It was assumed that these individuals belonged to the 45–49 age group. The number of susceptible individuals at the start of the epidemic was obtained from INE, the remaining initial conditions were set to 0. We fit the model until the 15th of January 2021.

In order to estimate the changes in disease transmission that occurred due to the implementation or lifting of NPIs, we divided the Portuguese COVID-19 epidemic in 6 periods. Each period starts and ends with a significant change in transmission:

1.  Starts at the 10th of February 2020 (day zero) and encompasses the first exponential growth phase of the epidemic, which was curbed by the implementation of the closure of schools and state-of-emergency NPIs.

2. Covers the first descendent incidence phase. Schools were closed and state-of-emergency was in effect. Ends with the phase out of the state of emergency during May-June. Schools were closed during this period.

3. This period covers the transition from the state-of-emergency to the summer. This period ends as the new exponential growth phase beginning and schools opening.

4. Covers the second exponential growth phase after the summer is over. During this period several softer NPIs were imposed during October and November. Schools are opened during this period.

5. This period takes into account the short window between the reduction in infectious contacts caused by the NPI implemented during October and November to the end of the year.

6. This period starts with the third exponential phase of the Portuguese epidemic during the Christmas holidays and continues until the 15th of January.

Although changes in the transmission are related to the implementation and lifting of NPIs, it is usually not expected that these take effect immediately. Hence, it is important to estimate when these changes occurred and what was their relative change in the transmission.

For the period prior to the epidemic, we assume the Portuguese baseline contact pattern as in [10]. In the following periods we assumed that this pattern changed. Meaning that matrix parameters $U^k$, $Y^k$, $W^k$ and $Z^k$, for $k = S, A$, changed in time. Thus, following a method similar to the one described in [9], we assumed these matrices to be of the form $\alpha(t)I$, where $I$ is the identity matrix and $\alpha_j(t)$ is a piecewise function of the form:

$$\alpha_j(t) = \begin{cases} 1 & , t < br_{j1} \\ \alpha_{ji} & , br_{ji} \leq t < br_{j(i+1)} \\ \alpha_{j5} & , t \geq br_{j5} \end{cases} \quad , i = 1, 2, 3, 4; \quad j = U^k, Y^k, W^k, Z^k, \quad (26)$$

where $br_{ji}$ is the moment in time where most likely occurred a change in disease transmission due to the occurrence $i$ in the setting $j$, that is, these parameters set the beginning and ending of an epidemic period and are calibrated in the model fitting procedure. The parameter $\alpha_{ji}$ represents the relative change in contacts. Since no data is available on the different contact patterns in the work, home and other locations settings, during the COVID-19 pandemic, we assumed the same function for the contact matrices of these locations. For the school contact matrix we assumed no contacts between the closure of schools starting in 16 March 2020 and ending in the 15 September. The same was assumed for the Christmas holidays. Upon opening, we assumed a reduction in the schools reflecting the measures implemented, such as, the use of face mask (47% effectiveness), the alternated schools schedules (33% less contacts) and overall compliance to mask usage (90%). The effectiveness of the mask usage was based on a meta analysis study which pooled eight studies that measured the reduction in infection risk associated with mask usage by non-health workers [17].

The model was numerically solved via the `lsoda` function in the `deSolve` package in the `R` language and environment for statistical computing, version 4.0.3 [18]. Model was fitted to the ICU prevalence data ($H_{ICU_{obs}}$) and fitting was performed via the differential evolution algorithm using the `R` package `DEoptim` [19] by minimising the square root of the sum of squares presented below:

$$\underset{\phi}{\text{argmin}} = \sqrt{\frac{1}{m} \sum_t (H_{ICU}(t, \phi) - H_{ICU_{obs}})^2},$$

where $H_{ICU_i}(t)$ represents the total number of ICU individuals at a time $t$ as given by the model, $m$ is the number of days considered in the fit and $\phi$ is the set of parameters we want to calibrate. We ran the differential evolution algorithm for 250 iterations, resulting in a

$\sqrt{MSE} = 27.998$. Figure 3 displays the model fit to the data, as well as, the hospitalised and death observed and model values.

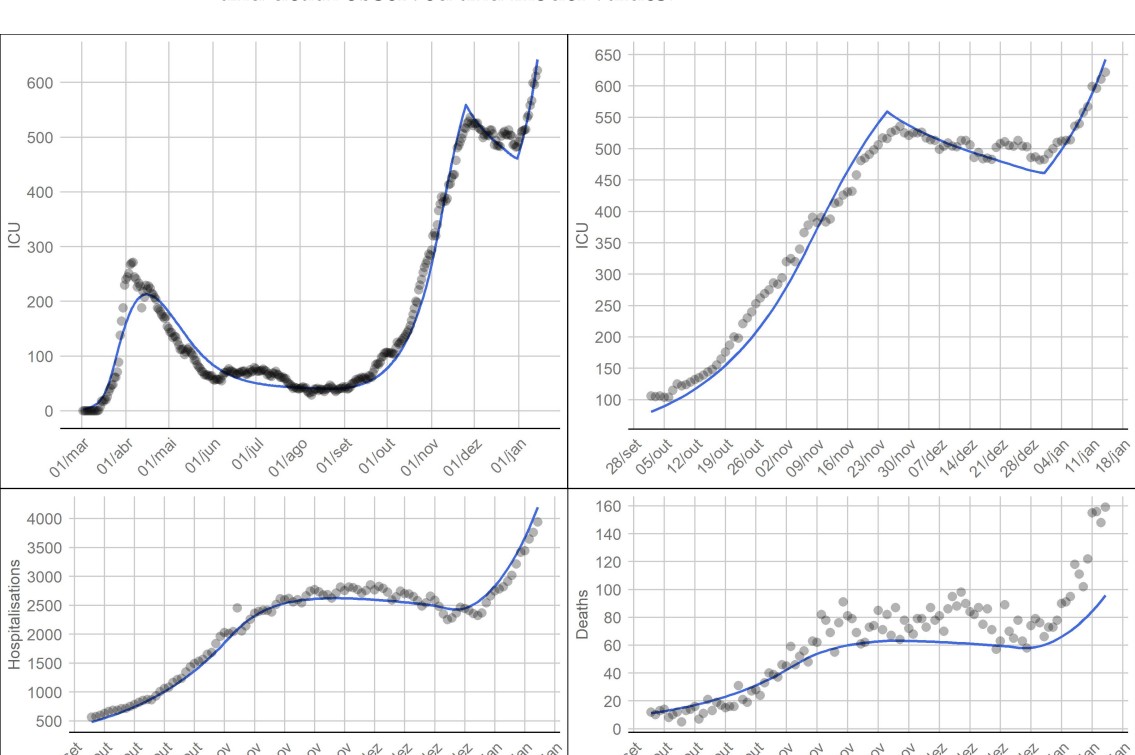

**Figure 3.** Model fit to ICU data since the start of the epidemic (**top left**). Model fit to ICU data from the 1st of October 2020 to the 15th of January 2021 (**top right**). Observed and model values for hospitalised individuals (**bottom left**) and deaths (**bottom right**). Solid lines represent the model, the dots represent observed values (data available in [14]).

The first epidemic period corresponds to the initial growth phase of incident cases with an effective reproductive number ranging from 2.22 to 2.5. On the 18th of March the model estimates a change in disease transmission, as seen on Figure 4, reducing overall contacts in work, home and other settings by 69% and bringing the effective reproduction number below 1. This change coincides with the announcement of the state of emergency, which began on the 22nd of March 2020, schools had been closed since the 16th of this month. This decrease in incident cases lasted until the 10th May. During this period, a gradual lifting of the lockdown measures started. The phase down of measures occurred on the 4th of May, 18th of May and 1st of June. The model estimates that on the 10th of May the effective reproduction number changed to 1.00, indicating an increase in transmission, probably due to the start of the phase out. This phase lasted for several months, where Portugal went into a low and steady incidence rate which in turn resulted in low hospitalisations, ICU cases and deaths.

By the end of the summer of 2020 Portugal saw an increase in the transmission. The model estimates that on the 18th of August there was an increase in the transmission ($\mathcal{R}(t) = 1.26$) which increased with the opening of schools on the 15th of September ($\mathcal{R}(t) = 1.35$). This increase in the transmission coincided with the end of the summer holidays and back to school and work periods. During this period there was also an increase in overall mobility of the population [20]. Finally the model estimates a reduction in infection transmission on the 2nd of November which coincides with the mitigation measures implemented during October and early November of 2020.

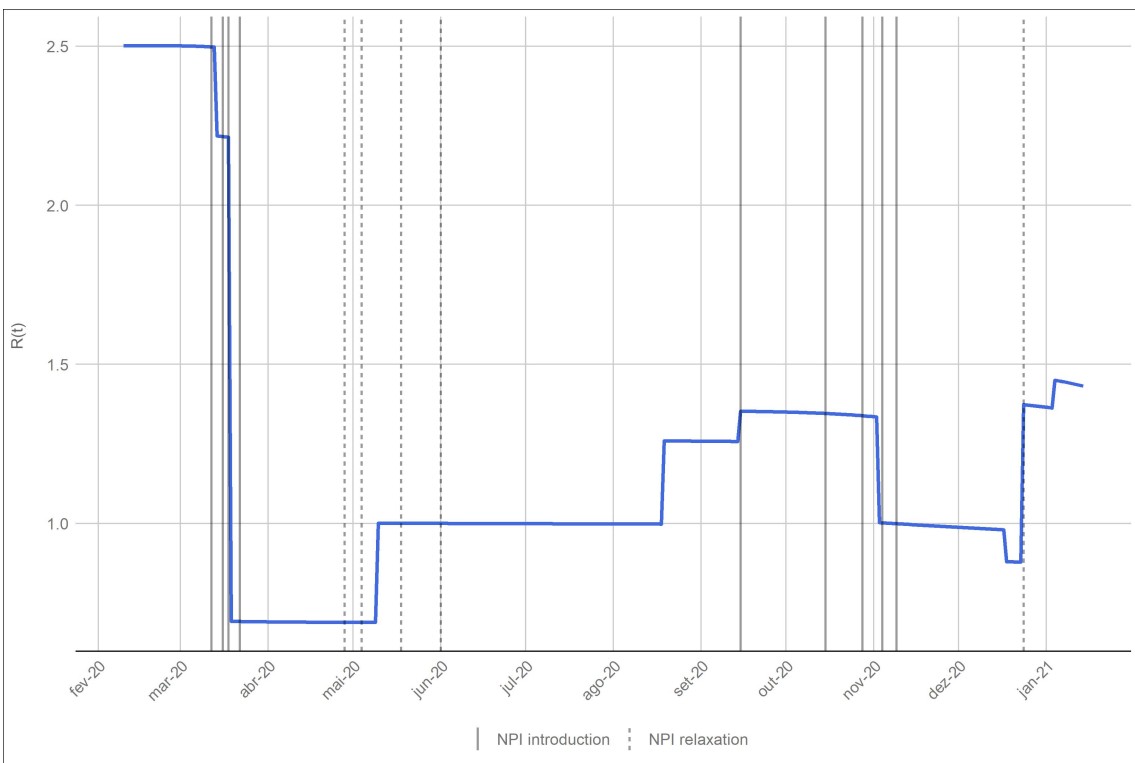

**Figure 4.** Evolution of the effective reproduction number given by the model. The solid vertical lines correspond to the date of the implementation of a NPI and the dashed lines mark the relaxation of the NPIs in place.

Schools closed for Christmas holidays on 18th of December. Although the number of hospitalised and ICU individuals was high, both were stable. On the 23rd of December the model estimates a significant change in transmission. The effective reproduction number goes from 0.88 to 1.37. It is important to note that this is probable due to the softening of gathering restrictions during this period. This transmission change was further exacerbated by the opening of schools on the 4th January 2021.

The model also estimates that approximately 8% of the population had been infected by the virus by 15 January 2021. This low number demonstrates the efficacy of the public health transmission prevention measures implemented so far, and also shows that the changes in the effective reproduction number were due to a change in contacts which were a reflection of the implemented measures. However, these results should be compared to seroprevalance data, mainly due to the uncertainty related to asymptomatic cases and its importance in the disease transmission dynamics.

On the 15th of January 2021 the Portuguese government instated a new state of emergency similar to the one implemented in March of 2020 in order to curb the high number of incident cases, hospitalisations and deaths related to COVID-19. Schools were also closed on the 22nd January. In order to evaluate the impact of these interventions and measure their future effect on ICU admissions we consider the following simulation: (a) we assumed that the lockdown and school closure had the desired effect in transmission reduction on the same day of their implementation; (b) we assumed that the reduction in contacts upon implementation to be the same as the one estimated for the first lockdown in March of the previous year; (c) we created scenarios for no interventions, 1 month, 45 days and 2 months of school closure plus lockdown; (d) we assumed that once the lockdown was over and schools reopened the increase in contacts was the one estimated for August of 2020. Figure 5 depicts this simulation.

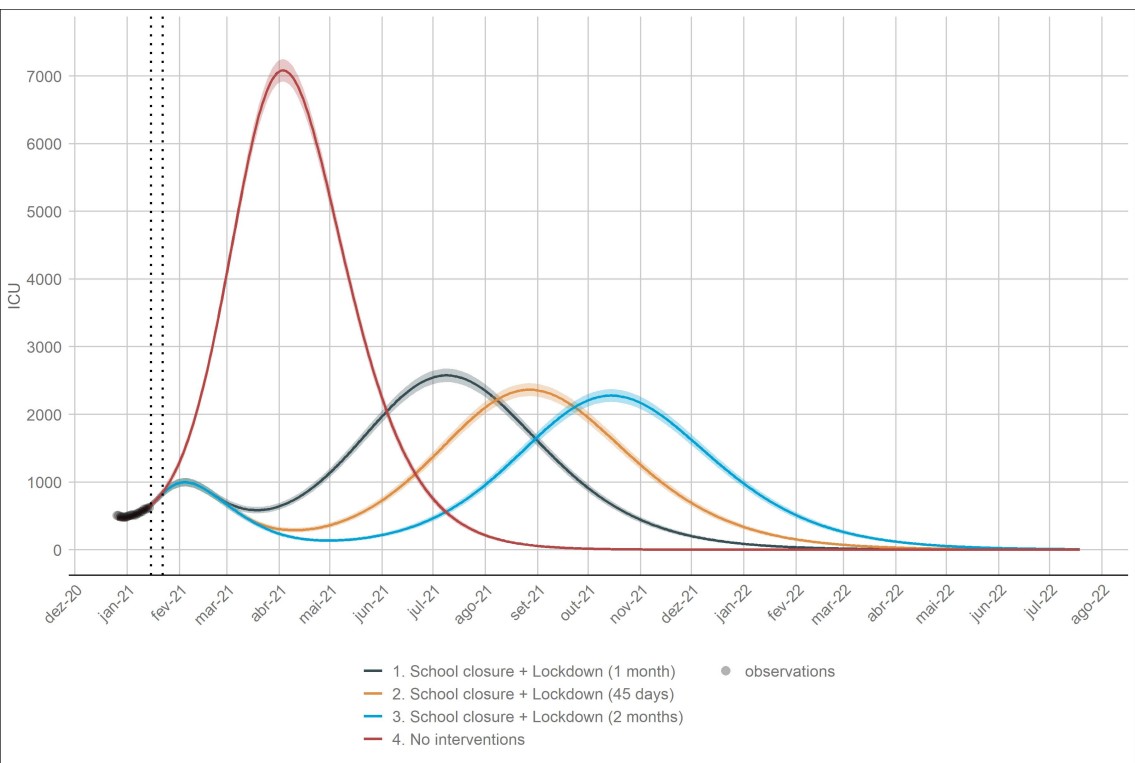

**Figure 5.** School closure and lockdown simulations on the number of ICU cases. The dotted vertical lines depict the implementation of the lockdown and the closure of schools. The simulation in red represents the scenario with no interventions.

The results show that the one month and 45 days scenarios are not enough to bring the number of individuals in ICU below 200. The two month scenario shows that this number is achieved in early April. It is important to note that the public health system is heavily burden by the disease, meaning that a short lockdown might not be enough to reduce this burden. All the scenarios show an increase in the number of ICU cases after the end of the lockdown. This is expected since the number of susceptible individuals in the population is high.

Uncertainty in these estimates was obtained by assuming that the number of prevalent cases in ICU followed a Negative Binomial distribution with mean equal to the number of cases given by the model and a dispersion parameter that was obtained via the maximum likelihood method.

## 5. Discussion

The model depicted a good fit to the data. It was able to ascertain the impact of the implementation of past NPIs in the disease transmission. The model also served as a tool to create simulation scenarios for the effect of possible new interventions and their duration. In this way being able to forecast proper control strategies.

However, there are several obstacles in ascertaining the proper parameter estimation. Firstly, we assume that by changing contacts in the school setting there is no interference in the contacts in the remaining settings. During the initial lockdown in March, it was observed that mobility changed significantly [20], especially in the household setting. However, individuals in households have contacts with the same individuals each day, something that cannot be considered in the model. Information on contact patterns during the pandemic is key to understanding which settings were most affected by the interventions, in order to properly create scenarios for future change in disease transmission. Secondly, it was not possible to obtain most of the data by age-group for the period considered, which resulted in not being able to estimate age-dependent model parameters. Hence it was assumed that individuals in the population only varied according to their mixing patterns and hospitalisation risk.

Some studies have reported that young children are less susceptible to infection [21]. However, no study in Portugal at the present time reports such findings. Hence, it is assumed in the model that all individuals are equally susceptible to infection. It is important to note that, if in fact children are less susceptible to infection, it will greatly affect the impact of school opening and closure. Thirdly, in the data fitting procedure we assumed that the number of prevalent individuals in ICU, reported each day corresponded to the number of ICU beds occupied the day before, which means that we assumed that there is a reporting delay of 1 day in this process.

We estimate that changes in the transmission occurred concurrently with either the implementation of new mitigation measures, the lifting of set measures or with an increase of population mobility [20]. After the schools closed and the implementation of a country-wide lockdown, the model estimates that the basic reproduction number dropped from 2.5 to 0.69, which indicates that the implemented measures reduced disease transmission by 72%. This was the result of the schools closing on the 16th of March and the mandatory 'stay-at-home' from 22nd of March to the beginning of May 2020. During this period, individuals were only allowed to leave home to work, whenever it could not be done from home. Similar measures were adopted in other countries which also saw a reduction in the number of new cases [13]. The phase out of the measures during May and June of 2020 saw an increase in transmission. We estimate that disease transmission increased after 10 May, though with an effective reproduction number of 1, indicating a steady number of incident cases. The end of the summer saw an increase in disease transmission up until November. This increase was probably a result of a reduction in risk perception of infection during the summer, the movement of the population during the holidays and the returning to school and work in September. Although with an $\mathcal{R}(t)$ much lower than the $R_0$, this increase happened slowly and steadily, as reported by [16], resulting in a second and higher wave of cases and consequently an increase in ICU cases as depicted in Figure 3. 15th of October, 4th and 9th of November saw measures implemented to fight this increase in transmission. We estimate that a significant reduction in infectious contacts occurred after 2 November. December 2020 saw an increase in transmission on the 23rd, which might result from the usual gatherings during the holidays and new year's eve. This led the effective reproduction rise to 1.45 in the beginning of January 2021, which in turn resulted in more than 3500 hospitalised individuals and more than 600 individuals in ICU by mid January.

The results presented in this paper are similar to those described in [16]. The model shows that the effective implementation of NPIs curbed the increase of new cases and consequently, hospitalisations, ICU cases and deaths. The model's simulations also indicate that a short lockdown might not be enough to reduce the disease burden on public health services. A combination of a longer lockdown with an effective vaccination strategy might reduce the number of hospitalisations and deaths and help maintain the effective reproduction number below 1.

In this paper, we do not address the use of the isolated portion of the model, that is we do not consider the case $q > 0$. These individuals can be considered to only have contacts within their household and therefore contribute to a less extent to the disease transmission. However, this parameter is highly dependent on the efforts of public health official to trace and isolate infected contacts or on the perception of individuals of the population to self isolate upon exposure, hence it is expected that this value is inversely proportional to the number of new cases. This parameter could be used in tandem with the contact matrices to hypothesise an ideal strategy for epidemic control, that is, when incidence is low we can remove stringent lockdown measure and reinforce contact tracing teams and promote self-isolation. This topic will be further studied as new data becomes available.

**Author Contributions:** Conceptualization, C.C., M.L.M., P.P. and B.N.; methodology, C.C., M.L.M. and P.P.; software development, C.C. and J.F.P.; validation, M.L.M. and P.P.; formal analysis C.C., P.P. and M.L.M.; data curation, C.C. and J.F.P.; visualisation, C.C.; writing original draft preparation, C.C. and J.F.P.; writing review and editing, C.C., P.P., M.L.M. and B.N.; supervision, B.N., M.L.M.;

project administration, B.N., M.L.M. All authors have read and agreed to the published version of the manuscript.

**Funding:** The authors acknowledge financial support from the Fundação para a Ciência e Tecnologia—FCT through project 692 2ª edição Research 4 covid, project name Projeção do Impacte das medidas Não-farmacológicas de Controlo e mitigação da epidemia de COVID-19 em Tempo ReaL (COVID-19 in-CTRL). The first author also acknowledges FCT within the PhD grants "DOCTORATES 4 COVID-19", number 2020.10172.BD. The second author also acknowledges FCT within projects UIDB/04621/2020 and UIDP/04621/2020. The third author also acknowledges FCT within the Strategic Project UIDB/00297/2020 (Centro de Matemática e Aplicações, FCT NOVA).

**Institutional Review Board Statement:** The study was approved by the Ethics Committee "Comissão de Ética para a Saúde" of Instituto Nacional de Saúde Doutor Ricardo Jorge (16/03/2021).

**Informed Consent Statement:** Not applicable.

**Data Availability Statement:** Publicly available datasets were analyzed in this study. This data can be found here: https://covid19.min-saude.pt/relatorio-de-situacao/, accessed on 1 February 2021.

**Conflicts of Interest:** The authors declare no conflict of interest.

## Abbreviations

The following abbreviations are used in this manuscript:

| | |
|---|---|
| NPI | Non Pharmaceutical Intervention |
| ICU | Intensive Care Units |

## Appendix A

In Table A1 we describe the model's parameters description, values and the source. Table A2 describes the moments and description of the lifting/introduction of NPIs.

**Table A1.** Description, value and source for the parameters used in the model.

| Parameter | Description | Value | Source |
|---|---|---|---|
| $\beta$ | transmission probability | 0.068 (taken from $R_0 = 2.5$) | [13] |
| $\varepsilon$ | latent period | 1/3.8 days$^{-1}$ | [9] |
| $r_s/r_a$ | infectious period | 1/3.4 days$^{-1}$ | [9] |
| $\theta$ | probability of hospitalisation (age-dependent) | 5.1% for 0–49 years; 10.11 % for 50–59 years; 21.99 % for 60–69 years; 40.00 % for 70+ years; | [12] |
| $p$ | proportion of asymptomatic | 44.5% | [3] |
| $\rho$ | rate of progression out of $H$ | 1/9 days$^{-1}$ | ACSS/SPMS |
| $\pi$ | fraction of hospitalized individuals progressing to ICU (time dependent) | 12.6% 7.9% from 11/25 to 12/31 11.2% from 12/31 onwards | ACSS/SPMS |
| $\tau$ | proportion of hospitalization deaths | 19.2% | ACSS/SPMS |
| $\omega$ | rate of progression out of ICU | 1/20 days$^{-1}$ | ACSS/SPMS |
| $\mu$ | proportion of ICU deaths | 26.7% | ACSS/SPMS |
| $\alpha_A$, $\alpha_{Aq}$ | asymptomatic reduction in transmission | 50% | [22] |

**Table A1.** *Cont.*

| Parameter | Description | Value | Source |
|---|---|---|---|
| $\alpha_S$, $\alpha_{Sq}$ | symptomatic reduction in transmission | 0% | assumed |
| $School_r$ | contact reduction in schools | 33% | assumed |
| $School_c$ | school mask use compliance | 90% | assumed |
| $Mask_{eff}$ | mask effectiveness in reducing transmission | 47% | [17] |
| $S_0$ | initial conditions for $S$ | [436,202 455,843 504,940 545,322 550,444 547,680 566,594 672,422 784,224 789,733 745,178 740,141 676,762 622,912 549,591 1,107,921]$^t$ | INE |
| $E.nt_0$ | initial condition for $E$ | [0 0 0 0 0 0 0 0 71 0 0 0 0 0 0 0]$^t$ | estimated |
| $br_{1_{H;W;O}}$ | change in transmission 1 | 2020-03-18 (t = 37) | estimated |
| $br_{2_{H;W;O}}$ | change in transmission 2 | 2020-05-10 (t = 90) | estimated |
| $br_{3_{H;W;O}}$ | change in transmission 3 | 2020-08-18 (t = 190) | estimated |
| $br_{4_{H;W;O}}$ | change in transmission 4 | 2020-11-02 (t = 266) | estimated |
| $br_{5_{H;W;O}}$ | change in transmission 5 | 2020-12-23 (t = 317) | estimated |
| $\alpha_{1_{H;W;O}}$ | change in contacts after $br_{1_{H;W;O}}$ | 69% | estimated |
| $\alpha_{2_{H;W;O}}$ | change in contacts after $br_{2_{H;W;O}}$ | 55% | estimated |
| $\alpha_{3_{H;W;O}}$ | change in contacts after $br_{3_{H;W;O}}$ | 43% | estimated |
| $\alpha_{4_{H;W;O}}$ | change in contacts after $br_{4_{H;W;O}}$ | 58% | estimated |
| $\alpha_{5_{H;W;O}}$ | change in contacts after $br_{5_{H;W;O}}$ | 35% | estimated |

**Table A2.** Introduction and lifting of NPI adopted in Portugal, dates and descriptions. Lockdown refers to a mandatory stay-at-home order. In Portugal this refers to a declaration of "state-of-emergency" by the Portuguese government to provide a response to a national crisis. This state allows the implementation of severe measures to fight disease spread. "state of contingency" refers to the introduction of milder measures and "state of calamity" corresponds to a state in between contingency and emergency.

| Date | Description |
|---|---|
| 2020-03-12 | announcement of schools closure |
| 2020-03-16 | closure of schools |
| 2020-03-18 | lockdown ("state-of-emergency") announcement |
| 2020-03-22 | lockdown goes into effect |
| 2020-04-28 | announcement of lockdown phase-out |
| 2020-05-04 | first wave of lockdown phase-out |
| 2020-05-18 | second wave of lockdown phase-out |
| 2020-06-01 | third wave of lockdown phase-out |
| 2020-09-15 | "state of contingency" goes into effect |
| 2020-10-15 | "state of calamity" goes into effect |
| 2020-10-28 | outdoor obligatory use of mask |
| 2020-11-04 | lockdown measures on weekends for counties above 480/100,000 incidence |
| 2020-11-09 | "state of emergency" |
| 2020-12-24 | relaxation of measures during Christmas |
| 2020-01-15 | lockdown ("state of emergency") |

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
