# Peer review of "Mathematical Modelling of the Impact of Non-Pharmacological Strategies to Control the COVID-19 Epidemic in Portugal"

_mathematics, doi:10.3390/math9101084_

Round 1

Reviewer 1 Report

This  very interesting paper provides a partial analysis of the impact of  contagion mitigation strategies employed in Portugal between March 2020 and January 2021. I have however many questions. Please, bear with me, I'm a beginner to COVID modelling!

--"lockdown imposed on the 15th of January 2020": should this be 2021?

The analysis may be divided in:  

  1. employing an appropriate  ``SEIR type compartmental model" -- I suggest adding this as keyword, since most of current literature would call this a  ``compartmental model" ; I understand SEIR is quicker to type, but having the full name between keywords could increase visibility
  2. Subdividing  each compartment in 4 locations and n=16 age classes
  3.  estimation of the  parameters. Is it true that the only age dependent parameter is theta. Can you add more detail on  the   location dependent parameters, and how they are estimated?  (Typo: In "we were also able to estimate the effect that the NPIs implemented up to January 15th 2021" should "implemented" be replaced by "achieved"?)
  4. The main equations needed are the last two (unnumbered) on bottom of pg. 3, concerning contact matrices. Please, provide precise references explaining them (hopefully both textbooks and papers available on internet, to help the readers). Please, indicate in the caption what  the figures 2, 3 represent, and where the data comes from.
  5.   fitting of the model, over  6 periods. It seems however that figure 4 displays 8-9 R(t). Some are constant over a period (I can guess why :) but some have a small slope. What am I missing?
  6.  solving the model  numerically  with R . Please, clarify  even more what must be input into the lsoda function and DEoptim, and what comes out :) Even better, provide the R programs (let us share ...)

Reviewer 2 Report

Dear Authors, 

please see the attached file with my minor comments. 

Reviewer 3 Report

In this paper, the authors discuss the effect of non-pharmacological infection control measures on the spread of new corona viruses using the extended SEIR with age structure. The authors also calculate the basic reproduction number using the next-generation approach. This is an interesting study, but I think their model has a few flaws, as follows.

Major
The parameter r_s is questionable. This parameter should be the rate of transition from the infected state, but it is odd that it also includes the transition to the hospitalized state. I think the model should be improved.

Minor
1)    Page 3, line 2: Isn't d a mistake for (1-d)?
2)    Eq (10): Isn't tau a mistake for tau rho?
3)    Page 3, line 3 from the bottom: I don't know what the diagonal matrices U^k, W^k and Z^k mean.
4)    Page 6, line 8: The authors used the condition that q=0, c=1, but if q=0, it does not depend on c. I think the condition c=1 is unnecessary.
5)    I understand eq.(24), but what assumption led to eq. (26) below it? 

Reviewer 4 Report

This paper addresses a very important topic of investigation, namely Mathematical modeling of the impact of non-pharmacological strategies to control the COVID-19 epidemic in Portugal. The modeling idea is interesting. Although the model is simplistic, it is acceptable due to a lack of reliable data. Since it is a new disease, there are not enough data sources yet, which is understandable. I would advise the author to not only focus on Portugal but also on other countries to develop a good model.

Round 2

Reviewer 3 Report

I recommend that it be published as is. 
But I was a little surprised to hear that the form of the model used by the authors was derived from previous study (Saunders-Hastings et al.), because the average infectious period (including hospitalization H and H_icu) is not $1/r_s$ but $1/r_s+\theta/\rho+\theta\pi/\omega$. This can't be helped, since most epidemiological modelers are not so good at mathematics.